# Evaluation of the Maturity and Gene Expression of Sheep Oocytes and Embryos Cultured in Media Supplemented with Marjoram *(Origanum vulgare*) Extract

**DOI:** 10.3390/genes13101844

**Published:** 2022-10-12

**Authors:** Mashael M. Alotaibi, Ahmad R. Alhimaidi, Muath Q. Al-Ghadi, Aiman A. Ammari, Nawal M. Al-Malahi

**Affiliations:** Zoology Department, College of Science, King Saud University, Riyadh 11451, Saudi Arabia

**Keywords:** marjoram (*Origanum vulgare*), sheep oocytes, in vitro maturation, in vitro fertilization, embryo development, and gene expression

## Abstract

This study aimed to evaluate the effects of marjoram extract on oocyte maturation and gene expression in sheep oocytes and embryos. The first experiment studied the effect of the extract as an antioxidant to improve the in vitro maturation media used for sheep oocytes; the oocytes were matured in a TCM199 medium supplemented with 1 or 10 µg/mL of marjoram extract or the control, 0 µg, for 24 hr. Then, the maturation was estimated, and the gene expression was measured by using qPCR. The second experiment studied the effect of the extract on the development of sheep embryos produced in vitro; the fertilized oocytes were cultured in a SOF medium supplemented with 1 or 10 µg/mL of marjoram extract or the control, 0 µg, for 7 days. Then, the gene expression was measured using qPCR. The results showed that the marjoram extract did not improve nuclear maturation or the blastocyst rate. There was a significant increase in the level of GDF-9 gene expression in mature oocytes in the treatment groups. An increase in the expression of BCL-2 and EGR-1 genes was observed for the blastocysts in the 10 µg/mL group. We concluded that the marjoram extract did not improve nuclear maturation, but it did affect the expression of some genes in sheep oocytes and embryos.

## 1. Introduction

The past 80 years have seen major innovations in assisted reproductive technologies (ARTs) that have boosted the efficiency and profits of farmers and agricultural companies. Several problems limit the production of embryos in the laboratory, and therefore, research is directed toward improving the various techniques and steps involved in the process of producing embryos in the laboratory. Many problems limit the success of embryo production in vitro, which may be due to the incomplete final maturation of the oocytes, a high percentage of polyspermy after in vitro fertilization, or the low growth rate of the embryos during in vitro development. However, the success of embryo production in the laboratory is largely related to the success of in vitro maturation [1]. The type of maturation medium or culture medium, the hormonal additives, the proteins that are necessary to complete the maturation of the oocytes and the development of the embryo outside the body, and oxidation, might be reasons for the lack of oocyte quality. All of these can also limit the success of embryo production in vitro [2]. Attempts have been made to overcome these problems to develop and improve the maturation medium, from the maturity of the oocytes to the development of embryos in the laboratory, by adding supplements to the medium of oocyte maturation, followed by fertilization and embryo development [3]. The accumulation of reactive oxygen species may damage the capacity of the cell [4]. There are many factors that contribute to the insufficient quality of oocytes in the laboratory. One of the most important factors is oxidative stress, which accelerates the onset of apoptosis, especially in laboratory conditions, and thus, the antioxidant compounds will protect the oocytes and embryos from the reactive oxygen species (ROS) during maturity and the early stages of development [5]. Several studies showed the important role of antioxidants, such as Jamnik et al., who showed that royal jelly secreted from the hypopharyngeal and mandibular glands of honeybees and used with different concentrations of 1, 2, and 5 g/L, decreased intracellular oxidation in yeast cells [6]. Nagai and Inoue found that royal jelly had high antioxidative activity and scavenging abilities against radicals, such as superoxide and hydroxyl radicals [7]. Another study showed that royal jelly would act as a medicinal food in the human body when hydrolyzed, possessing higher antioxidative and scavenging activity against active oxygen species [8].

Other studies showed the importance of adding antioxidant compounds to the medium to protect the oocytes and embryos from the reactive oxygen species (ROS) during maturity and the early stages of development. Olson and Seidel concluded that the addition of 100 mM of vitamin E to a culture medium of bovine embryos also protected the embryos from reactive oxygen species by causing more development embryos to early and expanded blastocysts. Additionally, the embryos were larger compared to the control group after their transfer and collection from recipient cows [9].

Guerin et al. showed the importance of adding an antioxidant compound to a culture medium to protect the embryos from ROS originating from the embryo’s metabolism and/or surroundings [10]. Urdaneta et al. illustrated the effects of adding 100 μM of cysteamine to an in vitro maturation (IVM) medium, which led to the increased embryo development of prepubertal goat oocytes [11], L-carnitine [12], melatonin [13], fenugreek seed extract [14], and green tea extract, improving the efficiency of oocytes and blastocyst development before implantation in different mammalian species [15].

The early expression of some genes may have a role in oocyte maturation and embryo development, such as the imbalanced expression of the survival gene BCL-2, and the death or apoptosis gene Bax, which is an important cause of the improper in vitro maturation of sheep oocytes [16]. Additionally, heat shock conditions increase the production of reactive oxygen species (ROS) in the fertilized oocytes of mice and embryos with two cells [17]. The balance of heat shock protein genes (HSPs) reduces the harmful effects of any type of stress [18]. In sheep, growth differentiation factor-9 (GDF-9) and the protein of bone morphogenetic promoting factor-15 (BMP-15) are both essential for normal follicle growth and oocytes [19,20]. They play a crucial role in determining the rate of ovulation [21,22] and preventing the cellular death of ovarian follicles [23,24].

At present, there is a growing interest in the natural antioxidants arising from medicinal and food plants, which may contribute to preventing oxidative damage. *O. vulgare* belongs to the Lamiaceae family, contains phytoestrogens, and is a medicinal plant used in traditional treatments for reproductive problems and women’s health [25]. It is known for its strong antimicrobial and antioxidant activities [26], which explains its use in traditional medicine. Marjoram (*O. vulgare*) is an aromatic plant that is a source of active bio-compounds and contains a high percentage of minerals, vitamins, and antioxidants, as well as high phytoestrogens, antifungals, antimicrobials, and phenolic acids [27,28].

Abshenas et al. showed that *O. vulgare* improved the viability and motility of sperm in mice after chronic administration of copper sulfate [29]. Another study indicated that the crude leaf extract of *O. vulgare* reduced the length and density of the blood vessels of the chick chorioallantoic membrane by acting as an antiangiogenic agent because it has antioxidant and anticancer effects [30]. The addition of *O. vulgare* extract to the maturation medium for sheep spermatozoa increased the number of capacitated sperm [26]. Mbaye et al. illustrated that in vitro supplementation with oregano essential oil improved the parameters for sperm mobility and can be used as a safe therapeutic alternative for managing motility dysfunction in asthenozoospermic patients [31].

Therefore, the present study hypothesizes that the addition of marjoram *O. vulgare* extract to the oocyte maturation medium might reduce oxidative stress, hence, increasing the efficiency of oocyte development and enhancing embryo development in vitro with early gene expression.

## 2. Materials and Methods

### 2.1. O. vulgare Plants and Extract Preparation

An *O. vulgare* plant was collected at the flowering phase from a farm in Medina city, Saudi Arabia. The collected plant sample was dried in a ventilated place, and the leaves were separated from the stem. The water extract of *O. vulgare* was prepared according to [32], where 50 g of leaves were added to 500 mL of distilled water and boiled for 2 h at 100 °C. The water was removed, and 300 mL of distilled water was added again and left to boil for one hour. The collected water, at both times, was filtered and dried at 40–50 °C under low pressure in a rotary evaporator, and then, the collected extract was stored at 4 °C until its use. Two stock solutions were prepared from the last collected extract: the first stock concentration was 5 mg/mL, and the second was 50 mg/mL. From each one, respectively, we obtained the final concentrations of 1 µg/mL and 10 µg/mL.

### 2.2. Ovary and Oocyte Collection

The ovaries were collected from the central slaughterhouse in Riyadh, Saudi Arabia, in a 0.9% NaCl saline solution and brought to the lab within 2–3 h at a temperature of 25–35 °C. The oocytes were aspirated from the ovarian follicles using a 20-gauge injection needle connected to an aspiration pump. The collection medium of follicular fluid consisted of phosphate buffer saline (PBS) solution, heparin (0.14 mg/mL), and bovine serum albumin (BSA, 4 mg/mL). Oocytes surrounded with two layers or more of cumulus cells were used.

### 2.3. Experimental Design

Two experiments were conducted as follows.

The first experiment assessed the nuclear maturation of the oocytes and measured the gene expression of matured sheep oocytes. This experiment included three groups: Group 1: oocytes cultured in a defined maturation medium as the control; group 2: oocytes cultured in a defined maturation medium + 1 µg/mL of marjoram extract (Treatment 1); group 3: oocytes cultured in a defined maturation medium + 10 µg/mL of marjoram extract (Treatment 2). There were three replicates for each group in each assessment.

The second experiment measured the gene expression of the sheep embryos produced in vitro. This experiment included three groups: (i) defined culture medium (control group); (ii) defined culture medium + 1 µg/mL of marjoram extract (Treatment 1); (iii) defined culture medium + 10 µg/mL of marjoram extract (Treatment 2). There were three replicates for each group.

For the first experiment, the oocytes were collected and then cultured in a 3.5 cm Petri dish containing a defined maturation medium consisting of TCM199 Earle’s salts (Sigma M4530) supplemented with fetal bovine serum FBS (Sigma F4135) 10%; sodium pyruvate (Sigma P5280) 0.5 μM, FSH (Sigma F8174) 0.02 IU/mL, LH (Sigma L5269) 0.023 IU/mL, estradiol 17β (Sigma 2758) 1 μg/mL; cystamine 100 M/mL and antibiotic gentamycin (Caisson ABL03) 50 μg/mL for the control group. For the two treatment groups, the *O. vulgare* water extract was added to the maturation medium at a concentration of 1 or 10 μg/mL. Then, the oocytes were cultured for 24 h at 38.5 °C and 5% carbon dioxide with a high humidity. The oocyte nuclear maturation was then assessed, and the gene expressions of heat shock gene 70 (HSP70), cell death resistance gene (BCL-2), growth factor and cellular differentiation gene 9 (GDF-9), receptor of cellular transformation factor 1 (TGFßR1), and early growth response-1 (EGR-1), were assessed in both the control and treatment groups to determine the suitability of this culture for oocyte maturation.

For the second experiment, sheep oocytes were applied to the dish in the same previous culture medium after 20–22 h; then, frozen or collected sperm via electro-ejaculator ram semen was used to fertilize the matured oocytes (having the first polar body). The fertilized oocytes were cultured externally in a synthetic oviduct fluid (SOF) culture medium (Caisson IVL05) containing BME amino acids 50 x (Sigma B 6766,) and MEM nonessential amino acids (100%) (Sigma M 7145), bovine serum albumin BSA (6 mg/mL) and gentamycin antibiotic at a concentration of 50 μg/mL for the control group. For the two treatment groups, *O. vulgare* extract was supplemented to the culture medium at a concentration of 1 or 10 µL/mL, and then, the embryos were incubated in chamber conditions of 5% CO_2_, 5% oxygen and 90% nitrogen for 7 days in the incubator. Then, the rates of embryo development of each cleavage stage, the rate of progression to the blastocyst stage, and the estimation of BCL-2, HSP70, GDF-9, TGFßR1, and EGR-1 gene expression in the developing embryos were recorded.

### 2.4. Evaluating the DNA in the Cultured Oocytes

The mature oocytes were washed in 100 μL drops of hyaluronidase (300 IU/mL) to remove the follicle cells around the oocytes and then washed in PBS. Later, the oocytes were placed in ethanol and iced acetic acid at a ratio of 1:3 for 24–48 h. The oocytes were pulled and placed on a glass slide and covered with a slide cover. Then, 0.1% aceto-orcein dye was inserted under the cover. After 10–15 min, the slide was examined under an optical microscope, and the chromosome was examined with a power of 1000X. The meiosis cell division stages in the oocytes were recorded as follows: germinal vesicle (GV), germinal vesicle breakdown (GVBD), meiosis I (MI), anaphase I, and meiosis II (MII). To estimate gene expression, the cumulus cells were removed from the mature oocytes by placing them in 100 μL drops of hyaluronidase 300 IU/mL for 2–3 min and then washing them in the physiological solution, 0.1% PBS-PVA. Later, 30 mature oocytes were collected for each of the three experimental groups, preserved in 5 μL of the physiological solution 0.1% PBS-PVA in a 1.5 mL Eppendorf tube, and frozen at −80 °C until RNA isolation.

### 2.5. Sperm Capacitation Medium Preparation

The sperm capacitation medium used in this trial consisted of HEPES medium (Caisson IVL01) with heparin 1 µg/mL and sodium pyruvate 0.5 mM, bovine serum BSA 6 mg/mL, and gentamycin antibiotic 50 μg/mL, and was then incubated for at least two hours before its use. Sperm was collected from rams via an electro-ejaculator. Then, 100 µL of semen was placed in a 15 mL tube containing a capacitation medium and incubated for 1–2 h. Later, 700–800 microliters of supernatant and 1 mL of the previous capacitation medium were added and centrifuged at 1500 rpm for five minutes. The suspension was discarded, and 1 mL of the deposit was added and centrifuged at 1500 rpm for five minutes. The suspension was removed and added to the 1 mL deposit from the capacitation medium and placed in the incubator for 30 min to increase the fertility capacity. The sperm concentration was determined using a haemocytometer.

### 2.6. In vitro Fertilization of Oocytes: Fertilization Medium Preparation

The fertilization medium consisted of an IVF-TL fertilization medium (Caisson IVL02) with 1 μg/mL heparin and sodium pyruvate 0.5 mM, BSA 6 mg/mL, and the antibiotic gentamycin 50 µg/mL. The fertilization medium was filtered using a 0.22 μm syringe filter. A 6 cm Petri dish containing 100 μL drops was prepared and covered with mineral oil for oocyte washing, and a 4-well tissue culture plate was prepared for culturing and covered with mineral oil. The fertilization medium was prepared at least two hours before the experiment and was incubated at 38.5 °C in a 5% CO_2_ and high-humidity incubator.

After 24 h from oocyte maturation, the oocytes were washed three times in the fertilization medium dish that was prepared. Then, the oocytes were cultured on the 4-well tissue culture plate that was prepared. Then, fifty microliters of treated sperm were added to the oocytes with the final concentration of 1 × 10^6^ sperm/mL and incubated for 2 h at 38.5 °C in a 5% CO_2_ and high-humidity incubator.

### 2.7. Assessment of Gene Expression

The RNA was extracted from the oocytes and embryos by a PureLink RNA Mini Kit (cat. no. 12183018A) according to the instructions in the analysis kit, summarized as follows:

The sample-keeping fluid was removed by centrifuging the samples at a speed of 4500 g for 5 min, extracting 3–4 μL of the liquid, and then adding 300 microliters of lysis buffer cell solution to each sample and blending with a vortex-shaking device. Following that, the RNA was reverse-transcribed into cDNA using the High Capacity cDNA Reverse Transcription Kit (Cat. No. 4368813) according to the kit’s instructions. Finally, using the Applied Biosystems ViiA 7 System Real-Time PCR/ThermoFisher Scientific, a reaction of 10 μL of SYBR Green Master Mix, 1 μL of forward and reverse primers, and 2 μL of cDNA was completed with 20 μL of nuclease-free water, and the gene expression was measured.

The following genes were detected: HSP70, BCL-2, GDF-9, TGFßR1, and EGR-1, which control oocyte maturation and the development of sheep embryos produced in vitro. The degree of differential expression of each gene under the different treatments was calculated by the comparative CT method [33] using the following equation:2ΔΔ. ct = [(CT of target gene − CT reference gene) sample A − (CT of target gene − CT of reference gene) sample B)]

Sample A: the treated group, Sample B: the control group.

The serial composition of the primers used to measure gene expression is shown in (Table 1).

### 2.8. Statistical Analysis

Three replicates were conducted for each trial group, and then the data were analyzed using the Statistical Package for the Social Sciences (SPSS Inc.), Chicago, IL, USA. One-way ANOVA was performed to identify the differences between the averages of the total oocytes in the stages of nuclear maturation: GV, GVBD, MI, anaphase I, and MII. The average rate of cleavage, embryo count, and blastocyst stage rate for each of the three experimental groups (0, 1, and 10 μg/mL) were calculated. We also analyzed the level of gene expression in the oocytes and embryos. Duncan’s test was conducted for the dimensional comparisons between the groups. A *p* < 0.05 was identified as the significance level in the statistical analysis, and data are represented as the average ± standard error [40]. 

## 3. Results

The rate of nuclear maturation of sheep oocytes and their arrival to MII in the *O. vulgare*-treated groups compared to the control group showed non-significant differences, except in the GVBD stage, and a non-significant difference in the rate of degenerated oocytes among the three groups (Table 2).

After an analysis of the results of gene expression during oocyte maturation, it was found that the addition of the extract to the in vitro maturation medium had an inhibitory effect on the BCL-2 and HSP70 gene expression of the two experimental groups compared to the control group at *p* < 0.05. It was observed that their gene expression decreased with the presence of the extract compared to that of the control group. There was a significant difference at *p* < 0.05, recorded in the GDF-9 gene expression in the two treated experimental groups compared to the control group, and a higher expression was observed at higher concentrations (10 µg/mL). For TGFRβ1 and EGR-1, there was a significant increase at *p* < 0.05 in the first treatment group (1 μg/mL) compared to the control group (Table 3).

The embryo cleavage stages are shown in (Figure 1) (from a zygote or 1-cell stage to the blastocyst), and the rate of access of embryos to the blastocyst stage was studied in oocytes that were matured and fertilized in vitro and then cultured in the medium supplemented with the extract of *O. vulgare*. The rate of embryo cleavage was calculated by the following equation:Cleavage rate = No. of cleaved embryos/No. of cultured embryos × 100

The average number of embryos at the morula stage was significantly lower in the second experimental group (10 μg/mL) than in the control and the first experimental (at 1 μg/mL) groups. While there was no significant rise in the rate of cleavage in the first group (1 μg/mL) compared to the second group (10 μg/mL), there was also no difference between the two treated groups and the control group in the embryo development rate. There was a non-significant increase in the blastocyst development rate of embryos in the second treatment (10 μg/mL) compared to those in the other two groups Table 4.

The gene expression in the blastocyst stage showed a significant increase at a *p* < 0.05 for the BCL-2 gene in the first treatment (1 μg/mL) compared to that of the control group, as well as in the second treatment (10 μg/mL), while there was no significant difference in the levels of gene expression of HSP70 and GDF-9 in the three groups. There was a significant increase in the TGFRβ1 and EGR-1 genes in the second trial treatment of 10 μg/mL compared to that of the other two groups (Table 5).

## 4. Discussion

To our knowledge, this is the first study that investigated whether the supplementation of marjoram extract affects the nuclear maturation and gene expression of oocytes and the development and gene expression of pre-implantation embryos.

The results of this study show that there was a non-significant increase in the development of in vitro-matured oocytes treated with the *O. vulgare* extract to the MII phase compared to the control group. This is consistent with the previous results [41], in which L-carnitine was used in the culture medium for oocyte maturation, and there was no effect on the rate of MII maturation, despite its antioxidant properties. However, there was a noticeable increase in the division ratio at 7.5 mM and 10 mM compared to that at the lower concentrations in this previous study, which may be an indication that L-carnitine has a significant effect on cytoplasmic maturity compared to nuclear maturity. Other studies also did not mention any increase in the oocyte maturation rates when cystamine was added for the maturation of pig [42], horse [43], buffalo [44], and cattle oocytes [25].

The average number of embryos at the morula stage in this study was significantly lower in the second trial group (10 μg/mL) than in the control and first experimental groups (1 μg/mL), which is consistent with the results of Barakat et al. (2014), where it was shown that the addition of a green tea extract at high concentrations resulted in a lower rate of embryos in the morula stage [15].

There was also a non-significant increase in the rate of embryo development to the blastocyst stage in the second treatment (10 μg/mL), which is consistent with the results of Ali et al. (2003) in which the SOD and catalase antioxidants were added to the embryo culture medium, and there was no positive effect on embryo development [45].

The culture medium conditions influence gene expression in oocytes and embryos. ROS can also affect the level of gene expression, either by decreasing or increasing it [46]. In the results of our study, there was a significant decrease in the expression level of BCL-2 in mature oocytes in the two treatment groups compared to the control group, although the *O. vulgare* water extract contained antioxidants that reduced the types of ROS in the oocyte culture medium. However, this can be explained by the fact that during oocyte maturation and the early stages of embryo development, these oocytes and embryos are susceptible to apoptosis of cells due to the in vitro conditions. This is consistent with the findings of Barakat et al., which revealed that when fenugreek seed extract was added to the maturation medium, it resulted in a significant decrease in BCL-2 in the treated groups compared to the control group [14]. Additionally, when the antioxidant L-carnitine was added to the sheep oocyte maturation medium in vitro, Bax gene expression increased much more than in the control group [41].

This shows that the antioxidants added to the oocyte maturation medium in vitro do not necessarily reduce the expression of the programmed death gene (Bax) in the oocytes. The concentrations of the *O. vulgare* extract (1 or 10 μg/mL) used in the trial may have also been insufficient to reduce the free radicals generated during maturity that affect the regular gene expression of BCL-2. For the HSP70 heat shock protein gene, there was a decrease in the level of gene expression of HSP70 in the groups treated with the *O. vulgare* extract compared to that in the control group, and the production of this protein increased due to the high level of stress occurring in vitro [47]. The expression of the gene encoding this protein protects cells during heat stress [48].

The levels of gene expression of the GDF-9 and TGFRβ1 receptors in our results were higher in the first treatment group (1 μg/mL) than in the control group. This is also in agreement with Barakat et al., who found that treatment with fenugreek seed extract at the same concentrations used in this experiment increased the GDF-9 and TGFR1 receptors significantly in comparison to the control group [14]. This is consistent with the increase in oocyte access to the MII phase in the two treatment groups compared to that in the control group. GDF-9 and BMP15 are effective indicators of oocyte maturation because they are produced from the oocytes themselves [49]. For the EGR-1 gene, there was a significant increase in the level of gene expression in mature oocytes in the first treatment group compared to that in the control group. The EGR-1 early growth response gene is considered to be associated with oocyte efficiency [50].

The gene expression results for the BCL-2 gene were high in the blastocysts grown in the culture medium supplemented with the marjoram extract compared to the control group, despite the low concentrations; however, it did make a difference to the level of gene expression. This agrees with most studies in which there was a significant increase in the level of gene expression for BCL-2 in the blastocysts resulting from oocytes that were matured in a culture medium supplemented with antioxidants [34]. Additionally, as mentioned, the same result occurred when the antioxidant L-carnitine was added to the culture medium after the fertilization of sheep oocytes in vitro [41].

The gene expression of EGR-1 was reported to decrease significantly in all blastocysts [51]. Robert et al. (2001) noted a significant decrease in the regulation of the EGR-1 mRNA levels in all in vitro-produced blastocysts compared to their *in vivo* counterparts [50]. According to our results, the gene expression of EGR-1 in the second experimental group (10 μg/mL of *O. vulgare*) increased in the blastocyst stage compared to that of the other two groups. This could indicate the effectiveness of the *O. vulgare* extract in the embryo-growing culture medium in vitro, where the concentration of the 10 μg/mL extract increased the level of gene expression related to the protein response to early growth in the blastocyst embryos produced in vitro.

For the GDF-9 gene, there was no difference in expression levels in the different groups. There was a significant increase in the level of gene expression of TGFβR1 in the developing blastocysts in the *O. vulgare* culture medium with 10 μg/mL. The GDF-9 protein levels are positively linked to nuclear maturity and embryo quality and negatively associated with the progesterone levels in follicular fluid in primates [52]. Therefore, further studies are required to investigate the potential effects of marjoram *O. vulgare* in a culture medium.

## 5. Conclusions

From the above results, we conclude that adding an aqueous extract of marjoram (*O. vulgare*) to the maturation medium of sheep oocytes with the mentioned concentrations did not improve nuclear maturation and did not improve the blastocyst rate, but it did have an effect on the gene expression in oocytes and embryos.

## Figures and Tables

**Figure 1 genes-13-01844-f001:**
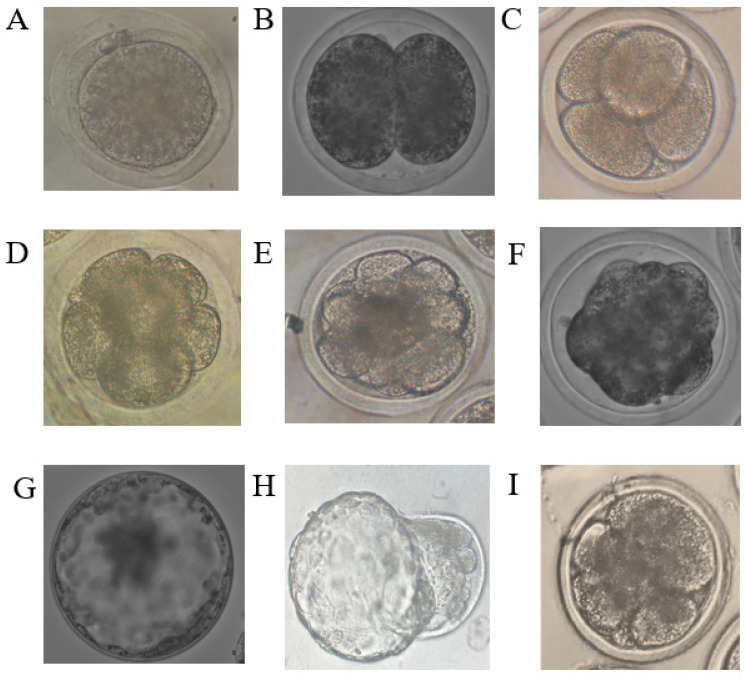
Images of embryos at different developmental stages obtained in the experiment. (**A**) = zygote, (**B**) = 2-cell stage, (**C**) = 4-cell stage, (**D**) = 8-cell stage, (**E**) = 16-cell stage, (**F**) = morula, (**G**) expanded blastocyst, (**H**) = hatching blastocyst, (**I**) = fragmented embryo.

**Table 1 genes-13-01844-t001:** The serial composition of the primers used to measure gene expression in sheep oocytes and embryos.

Function	Gene	Primer Sequence (5-3) Forward and Reverse	Fragment Size (bp)	GenBank Accession No.
Apoptosis	BCL-2	F:ATGTGTTGGAGAGCGTCA R:AGAGACAGCCAGGAGAAATC	182	NM 001166486.1 [34]
Stress	HSP70	F:AACATGAAGAGCGCCGTGGAGG R:GTTAACACCTGCTCCAGCTCC	171	NM 001267874.1 [35]
Nuclear maturation and embryonic development	GDF-9	F:TAGTCAGCTGAAGTGGGACA R:AGCCATCAGGCTCGATGGCC	224	AF078545 [36]
	TGFßR1	F:GGTTCCGTGAAGCAGAGATT R:GACACCAACCAG AGCTGAGTC	115	AY656799 [37]
	EGR-1	F:ACCAGTCCCAGCTCATCAA R:GAGCTCATCTGAGCGAGAGA	134	NM 001142506 [38]
Reference gene	GAPDH	F:ATGGGCGTGAACCACGAGAA R: ATGGCGTGGACAGTGGTCAT	146	NM 001190390 [39]

**Table 2 genes-13-01844-t002:** The effect of the addition of *O. vulgare* extract on the rate of nuclear maturation of sheep oocytes matured in vitro.

Conc. Of O.V.E	Total of Oocyte	Nuclear Maturation (Mean ± SEM)					Deg.
		GV	GVBDV	M I	Anaphase I	M II	
Control (0 µg/mL)	151	9.27 ± 0.02 ^a^ (14)	24.50 ± 0.03 ^a^ (37)	5.96 ± 0.01 ^a^ (9)	10.60 ± 0.02 ^a^ (16)	43.71 ± 0.04 ^a^ (66)	5.96 ± 0.01 ^a^ (9)
Treatment 1 (1 µg/mL)	152	13.16 ± 0.02 ^a^ (20)	11.84 ± 0.02 ^b^ (18)	5.26 ± 0.01 ^a^ (8)	13.82 ± 0.02 ^a^ (21)	47.37 ± 0.04 ^a^ (72)	8.55 ± 0.02 ^a^ (13)
Treatment 2 (10 µg/mL)	152	7.89 ± 0.02 ^a^ (12)	17.11 ± 0.03 ^ab^ (26)	4.61 ± 0.0 ^a^ (7)	9.87 ± 0.02 ^a^ (15)	56.61 ± 0.04 ^a^ (83)	5.92 ± 0.0 ^a^ (9)

O.V.E.: *O. vulgare* extract. The values are the mean ± S.E.M. GV: germinal vesicle, GVBD: germinal vesicle breakdown, MI: metaphase I, MII: metaphase II, Deg: Degenerated oocytes. (a,b) indicate significant differences within each column at *p*-value: (*p* < 0.05).

**Table 3 genes-13-01844-t003:** The effect of the *O. vulgare* extract on gene expression of in vitro-matured sheep oocytes.

Conc. Of O.V.E	Total of Oocyte	Nuclear Maturation )mean ± SEM(					Deg.
		GV	GVBDV	M I	Anaphase I	M II	
Control (0 µg/mL)	151	9.27 ± 0.02 ^a^ (14)	24.50 ± 0.03 ^a^ (37)	5.96 ± 0.01 ^a^ (9)	10.60 ± 0.02 ^a^ (16)	43.71 ± 0.04 ^a^ (66)	5.96 ± 0.01^a^ (9)
Treatment 1 (1 µg/mL)	152	13.16 ± 0.02 ^a^ (20)	11.84 ± 0.02 ^b^ (18)	5.26 ± 0.01 ^a^ (8)	13.82 ± 0.02 ^a^ (21)	47.37 ± 0.04 ^a^ (72)	8.55 ± 0.02 ^a^ (13)
Treatment 2 (10 µg/mL)	152	7.89 ± 0.02 ^a^ (12)	17.11 ± 0.03 ^ab^ (26)	4.61 ± 0.0 ^a^ (7)	9.87 ± 0.02 ^a^ (15)	56.61 ± 0.04 ^a^ (83)	5.92 ± 0.0 ^a^ (9)

O.V.E.: *O. vulgare* extract. The values are the mean ± S.E.M. (a,b) within each column and indicate significant differences (*p* < 0.05).

**Table 4 genes-13-01844-t004:** The effect of the *O. vulgare* extract on the numbers and proportions of sheep embryos developed in vitro.

Conc. of O.V.E	No, of Cutured Oocytes	One-Cell Stage No, Rate	2-Cell stage No, Rate	4-Cell Stage No, Rate	8-Cell Stage No, Rate	16-Cell Stage No, Rate	Morula Stage No, Rate	Cleavage No, Rate	Blastocyst No, Rate	Fragmented Embryos	Degenerated Oocytes
Control (0 µg/mL)	353	66 18.70± 0.02 ^a^	1 0.28 ± 0.05 ^a^	1 0.28 ± 0.05 ^a^	3 0.85 ± 0.04 ^a^	6 1.70 ± 0.06 ^a^	78 22.10 ± 0.02 ^a^	108 30.59 ± 0.02 ^ab^	19 17.59 ± 0.04 ^a^	105 29.75 ± 0.02 ^a^	74 20.96 ± 0.02 ^b^
Treatment 1 (1 µg/mL)	341	64 19.35± 0.02 ^a^	2 0.59 ± 0.07 ^a^	1 0.29 ± 0.05 ^a^	1 0.29 ± 0.02 ^a^	4 1.17 ± 0.05 ^a^	84 25.22 ± 0.02 ^a^	106 31.08 ± 0.03 ^a^	16 14.95 ± 0.03 ^a^	104 30.50 ± 0.02 ^a^	65 19.65 ± 0.02 ^b^
Treatment 2 (10 µg/mL)	257	50 19.46 ± 0.02 ^a^	0 0 ± 0.00 ^a^	1 0.39 ± 0.06 ^a^	1 0.39 ± 0.03 ^a^	6 2.33 ± 0.09 ^a^	38 14.79± 0.02 ^b^	60 23.34 ± 0.03 ^b^	14 23.33 ± 0.05 ^a^	72 28.02 ± 0.02 ^a^	75 29.18 ± 0.02 ^a^

O.V.E.: *O. vulgare* extract. Values represent the rate ± S.E.M. (a,b) within each column and indicate significant differences (*p* < 0.05).

**Table 5 genes-13-01844-t005:** The effect of the *O. vulgare* extract on the gene expression during cleavage of sheep embryos in vitro.

	The Fold Change in Gene Expression				
Conc. Of O.V.E	BCL2	HSP70	GDF9	TGFRβ1	EGR1
Control (0 µg/mL)	1.05 ± 0.20 ^c^	1.32 ± 0.70 ^a^	1.01 ± 0.12 ^a^	1.06 ± 0.26 ^b^	1.15 ± 0.36 ^b^
Treatment 1 (1 µg/mL)	9.57 ± 0.76 ^a^	0.61 ± 0.57 ^a^	0.41 ± 0.02 ^a^	1.36 ± 0.06 ^b^	1.82 ± 0.79 ^b^
Treatment 2 (10 µg/mL)	4.21 ± 0.48 ^b^	0.48 ± 0.06 ^a^	0.62 ± 0.36 ^a^	2.48 ± 0.13 ^a^	8.05 ± 2.84 ^a^

O.V.E.: *O. vulgare* extract. The values represent the mean ± S.E.M. (a, b and c) within each column and indicate significant differences (*p* < 0.05).

## Data Availability

Not applicable.

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
