# Peer review of "Evaluation of the Maturity and Gene Expression of Sheep Oocytes and Embryos Cultured in Media Supplemented with Marjoram (Origanum vulgare) Extract"

_genes, 2022, doi:10.3390/genes13101844_

Round 1
Reviewer 1 Report
Thank you for the opportunity to review this manuscript titled “Evaluation of the maturity and gene expression of sheep oocytes and embryos cultured in media supplemented with majoram (Origanum vulgare) extract”.
Comments follow:
TITLE
· The title is appropriate.
ABSTRACT:
Line 17: “marjoram extract increased the nuclear maturation rate”? This contradicts the conclusion of the abstract and the whole study. Very confusing.
The abstract does not state the effect of the said supplementation on embryo development.
INTRODUCTION
Overall, the introduction includes relevant information about the research conducted. However, authors must do substantial revisions as mentioned below.
Line 34-37
“Examples of this type of maturation and development culture medium, hormonal additives, and proteins are necessary to complete the maturation of oocytes and the growth of embryos outside the body, and it is believed that oxidation might be a cause of poor oocyte quality”
What do you mean by “examples of this type of maturation”?
Authors need to restructure this sentence since its not clear the message given. Combining “oxidation might be a cause of poor oocyte quality” with the rest of the sentence has no use. Keep it as a separate sentence.
Line 37-39
“Several attempts have been made to overcome these problems to develop and improve the maturation medium, from the maturity of the oocytes to the development of embryos in the laboratory, by supplementation of the maturation medium, followed by fertilization and embryo development”
Once again, this sentence also must be re-structured. Authors may obtain language support in this regard.
Line 46-53: Long sentence with over 90 words are not recommended in scientific writing.
Please break the sentence into several sentences so the reader can comprehend the message.
Moreover, authors may use the author’s name (such as Jamnik et al., as appropriate) when mentioning the author followed by a reference number at the end of the sentence.
Line 54-65: Same problem once again. Please break them into small sentences to increase clarity. Otherwise, the general reader will skip these sentences. Also, some grammatical errors were noted.
Line 66-67:
“So the role of antioxidants in the efficiency of oocyte or embryo production is controversial [16].”
Why do you start the sentence with “So”? The previous long sentence mentions only the positive aspects of antioxidants. If citation 16 mentions a negative or no use of anti-oxidants, then state what it is first. Subsequently, state the sentence in lines 66-67.
The word “so” is not standard in scientific writing. Please replace it with “Therefore” or other similar words.
Line 72-72
What is meant by “balance of heat shock protein genes”? not clear/
Line 86: “[32] showed that”. Please start the sentence with the author’s name rather than the in-text citation number. Ex. Jamnik et al. showed that. Add the in-text citation number at the end of the sentence.
MATERIALS AND METHODS
Line 103: “The water extract of (O. vulgare)” – correct it as “The water extract of O. vulgare”
Line 106-107: You got a powder form as an extract? Or what?
Line 108 - 1st should be corrected as 1st
Line 109-110: “From each one respectively we got the final concentrations ( 1 μg/ml, and 10 μg/ml )”
How did you get these final concentrations? By dilutions? State how it was achieved.
Line 113-114 : “at a temperature of 25-35 °C”. This is a huge temperature range which is absolutely not good for oocytes. Is this the case? This itself must have made a huge effect on the outcome of the whole study.
Line 114-115: “The oocytes were collected from the ovaries in an aspiration-like way”- please paraphrase this. Suggest the following: Oocytes were aspirated from the ovarian follicles using a 20-gauge injection needle connected to an aspiration pump. The collection medium of follicular fluid consisted of phosphate buffer saline (PBS) solution, heparin 0.14 mg/ml and bovine serum albumin (BSA, 4 mg/ml).
Line 116-117: “Two layers of cumulus cells surrounded the collected oocytes, and more were used only if needed”. Please re-write this sentence. Did you take only the oocytes with 2 layers of cumulus cells? Or what?
Line 120: “The first experiment assessed nuclear maturation”. Correct it as “The first experiment assessed the nuclear maturation”
Line 131: “For the 1st experiment, the oocytes were extruded and”. “Oocytes were extruded” is not appropriate. You may re-word this.
Line 134: 17ß – The symbol of beta is not clear in the version I reviewed. Please correct it.
Line 139-140 – “BCL2-HSP70-GDF9-TGFßR1-EGR1”. Please separate them with commas. Define these genes as this is their first appearance in the text.
Line 143-144: “electro ejaculator ram semen” . Correct it as ejaculated ram semen.
Line 144- “fertilize the matured oocytes with the first polar body”. Correct this as follows. “Fertilize the matured oocytes (having the first polar body). Otherwise, it is misleading.
Did you do a fertilization check? Did you use all the oocytes for subsequent culture or presumptive zygotes only?
Line 149: the (O. vulgare) water extract? Better to state “ O. vulgare extract was supplemented.
Line 150: CO2 to be correct as CO2.
Line 152: at each cleavage stage? There is only one cleavage stage. Please avoid the word ‘each’.
Line 158: The word alcohol is not needed. Ethanol is alcohol
Line 158: 1:3%? Not sure what the percentage means here.
Line 164: The word aggregate is not needed.
Line 164-165 – So when removing cumulus cells, no physical measures such as the use of a stripper are needed?
Line 171: heparin/ml? pyruvate 0.5 mm?
Line 172: gentamycin antibiotic 50 gμ/ml? units?
Line 179-180 - “The concentration of sperm per ml was confirmed by a hemocytometer red blood cell meter”. Correct is as follows. The sperm concentration was determined using a haemocytometer.
Line 185: “covered with mineral oil for washing”. What does this mean? Washing of what?
CO2 should be correctly written. Found the error in several places.
Line 188-189- “After 24 hours, it was taken from the incubator and washed”. What was taken from the incubator?
Line 190-191: Fifty microliters of treated sperm was 190 added to a final concentration of 1×106 sperm/ml and incubated for 2 h at 38.5 °C
Line 191-192: Please correct 1×106, CO2
How many oocytes were there per group during IVF?
Line 194-203: Please break this long sentence into small sentences. Then please check the grammar in the corrected paragraph. Long sentences are likely to cause more grammatical errors.
Line 216: Three iterations?
RESULTS
Line 227-229: “The rate of nuclear maturation of sheep oocytes and their arrival to MII in treated groups compared to the control group, additionally the difference in the rate of degenerated oocytes among the three groups illustrated in (Table 2)”.
You may correct the sentence as follows
The rate of nuclear maturation of sheep oocytes and their arrival to MII in O. vulgare treated groups compared to the control group, and also the difference in the rate of degenerated oocytes among the three groups are illustrated in Table 2.
Table 2
There is no consistency when using capitals.
Conc. Of O.V.E vs Total of Oocyts
Same inconsistency for bolt words
|
GV |
GVBD |
M I |
Anaphase I |
M II |
The following is not clear. “(a, b) indicate significant differences within each column at p-value: (P < 0.05)”. Does it mean different letters in each column indicates statistical differences?
Table 3
What are these values? Are they the fold change? Please specify in the table
“(1.oo ± 0.03)b” – please replace the letters with numbers.
Line 239-240: “from a zygote or 1-cell stage to the blastula” – Is this up to the blastocysts or blastula?
Line 240: the rate of access of embryos to the blastula stage. You mean rate of development of embryos?
Line 244: “Sector rate”. What does this mean? Isn’t this the cleavage rate?
The following is not clear. “(a, b) indicate significant differences within each column at p-value: (P < 0.05)”. Does it mean different letters in each column indicates statistical differences?
Table 4
Overall, the table is congested and busy.
The authors are having multiple evaluation points which may have been detrimental to the embryo development due to the frequent taking out of the laboratory.
How did you check the one-cell stage? Is this the fertilization check?
Authors must state clearly how they evaluated the embryos and at which time points of embryo culture. Were the time points of evaluations consistent?
The following is not clear. “(a, b) indicate significant differences within each column at p-value: (P < 0.05)”. Does it mean different letters in each column indicates statistical differences?
Table 5
|
BCL2 |
HSP70 |
GDF9 |
TGFRβ1 |
EGR1 |
Gene names above- make all of them bold
The following is not clear. “(a, b) indicate significant differences within each column at p-value: (P < 0.05)”. Does it mean different letters in each column indicates statistical differences?
DISCUSSION
Results have been discussed well. However, there is a higher number of amendments needed to make to improve the discussion as mentioned below.
Line 278-280: “To our knowledge, this study is the first investigation to indicates if the marjoram extract has effects on oocyte maturation and gene expression of oocytes and embryos with the mentioned concentrations or not”.
Suggest changing as follows
To our knowledge, this is the first study that investigated if the supplementation of marjoram extract has effects on nuclear maturation and gene expression of oocytes, and the development and gene expression of preimplantation embryos.
Line 280-282: This content does not fit here at the beginning of the discussion.
Line 285: The way the authors refer to previous studies is not agreeable. Ex . “which is consistent with the results of [15] found that the”. Authors need to pay more attention when constructing sentences. Here authors start another sentence before finishing one. They may re-write is as follows.
“which is consistent with the results of Barakat et al. (2014), where it was shown that………..”
Line 298: “embryo access to blastocysts”. Suggest “embryo development to blastocyst stage
Line 302-303: “Gene expression in oocytes and embryos is affected by the culture medium conditions and the levels of gene expression in oocyte and embryo development [45].” This sentence does not make sense. Please re-read and correct it.
Line 304-305: “In the results of our study” – suggest the following. According to the results of our study,
Line 307: (O. vulgare) water extract. Remove the brackets. Otherwise, its confusing when the main stem says water extract.
Line 310-311: “This is consistent with the results [46] when fenugreek seed extract was used”. The way authors have referenced previous studies and constructed sentences are wrong. Please refer to standard manuscripts and see how it is done. Please read my previous comments.
Line 310-315: Beak this sentence in to 2-3 sentences. Otherwise, use correct grammar.
Line 318: The concentrations of (O. vulgare)? Why the use brackets?
Line 326-328: “This is also in agreement with [46], where there was a significant rise of the GDF9 and TGFRβ1 receptors under treatment with fenugreek seed extract with the same concentrations used in this experiment compared to the control group” Please see if this sentence is readable? Please re-write.
Line 313: Please pay much attention to commas. They are missing in required places.
Line 336-338: “The expression of the BCL2 cell resistance gene was high in developing blastocyst culture medium supplemented with (O. vulgare) extract compared to that in the control group, despite its low concentrations, which made a difference in the level of gene expression”
It implies that BCL2 was tested in the blastocyst coulture medium which is not the case
“Despite its low concentrations” What do you mean?
Line 342: Robert and others (2001). Better to use “ Robert et al. (2001). Strangely, this kind of in-text citation was not observed previously in the text. This method is better compared to inserting a citation number in the middle of a sentence.
Line 345-346:
In our results – replace with “according to our results”
a second experimental treatment: replace with “2nd experimental group ((10 μg/ml of O. vulgare)
Please pay attention to units. Ex. (10 μ/ml)???
Line 352-353: transforming 352 fibroblast growth factor-beta recaptor-1 TGFβR1. Use brackets for TGFβR1
Line 356: fluid [53] in primates. Why is the citation number in the middle?
Line 356-357: “Therefore, further studies are needed to investigate the potential 356 antibacterial effects of marjoram (O. vulgare) in culture medium”. How come testing antibacterial properties came in the current context?
Conclusion:
· Conclusions have been drawn from the results.
References:
Reference numbers have been added twice. Please correct it.
Author Response
Thank sir, for all your comments...
I modified most of them within the manuscript and wrote ''done'' for the comments within the responses to you in a word file, and for some of them I wrote a comment within the same file.
Thank you
Nawal Al Malahi

Reviewer 2 Report
Dear authors,
The Genes journal is focused on genetic, but in your paper this part is not main. Gene expression analysis is too tight, not described affected metabolic pathways and other. Also, using of organic extracts for improving oocytes maturation is controversial issue because contain and biologicals effects of these extracts are unstable and hard prognosing.
I will not recommend your manuscript for publication, but if respected editors have a milder opinion, I has some questions for edition.
L113 – why you are didn’t use transport containers with stable 37 C temperature?
L120 – nuclear maturation or oocyte maturation?
L136 – what is the volume of dish is used? And what is volume of water solution of antioxidant was added? Is it no influence on medium osmolarity? Did you are control osmolarity?
L139 – not clear why selected this set of genes?
L191 – correct concentration 1*106.
L225 – check spelling in table header
L229 and next –poor tables analysis. Your must describe tables data for next using it in discussion, but text in Results mostly contains repeats of tables headers.
L260 – it is not necessary to illustrate standard morphology of oocytes and embryos, it is not aim of your study.
L271 – it is not clear what is Mean? In first line and column – 64 is absolute count. 18.7 – mean of what? Is it percent? What is SEM in this situation?
L277 – in Discussion compare your results with other research of influence of organic extracts on oocytes and gene expression.
Regards,
Author Response

(The authors gave the same response as above.)

Round 2
Reviewer 1 Report
The authors have revised the manuscript substantially. Hence, the manuscript has been improved and is in better shape to publish.
Reviewer 2 Report
Dear authors,
I still think your manuscript having some problems with consist. But if respected editors confirm your publication, I will confirm it too.
Best regards,